ecology

spiders, carabids, salinity, northwest France, species richness, traits

**Author for correspondence:**
Aurélien Ridel
e-mail: ridelaurelien@gmail.com

# Habitat filtering differentially modulates phylogenetic and functional diversity relationships between predatory arthropods

Aurélien Ridel[1,†], Denis Lafage[1,†], Pierre Devogel[1], Thomas Lacoue-Labarthe[2] and Julien Pétillon[1]

[1]UMR CNRS 6553 Ecobio, Université de Rennes, 263 Avenue du Gal Leclerc, CS 74205, 35042 Rennes cedex, France
[2]UMR CNRS 7266 LIENSs, Université de La Rochelle, 2 Rue Olympe de Gouges, 17000 La Rochelle, France

AR, 0000-0001-9048-1212

Mechanisms underlying biological diversities at different scales have received significant attention over the last decades. The hypothesis of whether local abiotic factors, driving functional and phylogenetic diversities, can differ among taxa of arthropods remains under-investigated. In this study, we compared correlations and drivers of functional diversity (FD) and phylogenetic diversity (PD) between spiders and carabids, two dominant taxa of ground-dwelling arthropods in salt marshes. Both taxa exhibited high correlation between FD and PD; the correlation was even higher in carabids, probably owing to their lower species richness. Analyses using structural equation modelling highlighted that FD and PD were positively linked to taxonomic diversity (TD) in both taxa; however, abiotic factors driving the FD and PD differed between spiders and carabids. Salinity particularly drove the TD of carabids, but not that of spiders, suggesting that spiders are phenotypically more plastic and less selected by this factor. Conversely, PD was influenced by salinity in spiders, but not in carabids. This result can be attributed to the different evolutionary history and colonization process of salt marshes between the two model taxa. Finally, our study highlights that, in taxa occupying the same niche in a constrained habitat, FD and PD can have different drivers, and thereby different filtering mechanisms.

†These authors contributed equally to this study.

# 1. Introduction

Description of spatial patterns of species assemblages is an objective of community ecology that can be directly used for biological conservation [1,2]. The study of factors driving local diversity is an essential step to understanding these patterns, and has long been performed using taxonomic diversity (TD) only. This approach does not consider all facets of biodiversity, such as accumulated evolutionary history traits that can be highlighted through phylogenetic diversity (PD) [3] or the diversity of morphological, physiological and ecological traits of an assemblage that can be revealed by functional diversity (FD) [4,5]. It is therefore important to study TD, PD and FD together for better understanding of the composition and dynamics of species assemblages [3], or even to set up priorities for biodiversity conservation in a fairer way [6]. In a complementary manner, TD provides information about the species composition of an ecosystem resulting from several processes such as habitat filtering or interspecific competition, PD highlights a part of the processes by providing information on the evolutionary relationships among coexisting species [3], when FD can reflect the differences of traits linking biodiversity, ecosystem functions and environmental constraints [7], as well as the functional response of species assemblages to environmental filtering [8].

Despite the fact that these metrics are seen to be complementary, their mutual relationships remain unclear [1], yet studying them is necessary to better understand all forces driving biodiversity patterns. Hypothetically, a positive correlation between TD and PD or FD is expected because the presence of more species can indirectly capture more lineage and functional traits. However, it has been shown that assemblages with similar numbers of species can have different values of PD and/or FD [9,10]. Moreover, the strength of a correlation between TD and PD depends on the time of evolutionary history of a given community, and can be influenced by other parameters, such as the symmetry of phylogenetic trees, length of branches, pool size of species and spatial autocorrelation [10]. Additionally, PD is often seen as a proxy for FD because the functional traits of an assemblage indirectly reflect its evolutionary history [11]. If traits are phylogenetically conserved, PD can also provide information about unmeasured functional traits [12], because in this case, PD results from the addition of all functional changes that occurred in the past. Some studies revealed a fluctuating relationship between PD and FD [1,13], which may depend on the pool size of species studied [14]. Furthermore, the relationship between PD and FD can also depend on the shape of the phylogenetic tree [15] and number of used functional traits. Finally, the inclusion of TD in both PD and FD calculations can lead to a correlation between them owing to a mathematical correlation artefact caused by a side effect [7,16–18]. A positive correlation between both TD and PD–FD is then expected as a rule [9].

To have a better understanding of the relationships between diversity metrics, it is important to understand what the drivers of these metrics are, and how they affect their relationships. Moreover, highlighting these drivers can improve the understanding of ecosystem functioning, as well as the observed biodiversity patterns, across all components of biodiversity.

Since the drivers of TD have been studied for a long time, elucidating the influence of factors driving PD and FD, which is a more recent challenge, is necessary [19–21]. While there is a large number of studies dealing with the taxonomic, phylogenetic and functional facets of diversity in plants (see a recent state of the art in [22]), there is a lack of knowledge about PD and FD drivers for less studied taxa such as terrestrial arthropods [21], especially by comparing taxa with similar ecological niches.

Here, we propose to carry out a multi-taxa approach considering all facets of biodiversity in salt marshes, a highly constrained environment. Salt marshes are transitional ecosystems between marine and terrestrial systems [23]. Owing to their intertidal position, salt marshes are subject to several environmental stressors, including periodic flooding and the resulting salinity gradient. These stresses have a strong impact on salt-marsh organisms [24], and most of the species found in these ecosystems have a high phenotypic plasticity, or even morphological, physiological or behavioural adaptations to cope with the stresses [25–27]. Among these organisms, terrestrial arthropods constitute the most diverse and abundant group in salt marshes [28,29], particularly spiders and carabids, which are dominant predatory arthropods in this habitat [30,31]. In addition, these two taxa play important functional roles in the environments where they act as both prey and predators (for carabids and spiders, see [32] and [33], respectively). To the best of our knowledge, the drivers of PD and FD have never been assessed and compared between spiders and carabids in salt marshes, which we did in this study by testing the following hypotheses.

Hypothesis 1: we expect the correlative relationship between PD and FD to be (i) positive because
    functional traits are usually phylogenetically conserved in constrained habitats

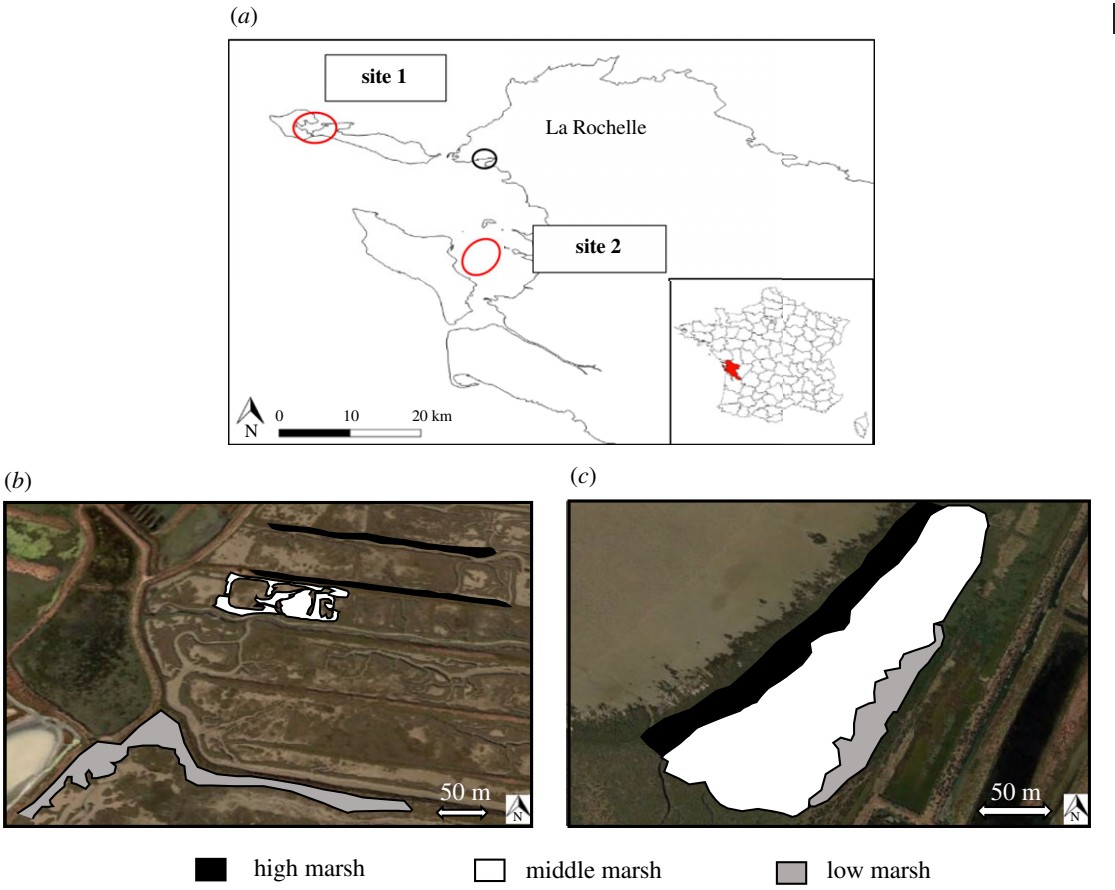

**Figure 1.** Location of study sites within the Charente-Maritime Department (western France). The three salt-marsh zones were defined on the basis of plant composition on the first site (*b*) and the second site (*c*). HM, high marsh (black); MM, middle marsh (white); LM, low marsh (grey). (*a*) Site 1 is located on the Ile de Ré, and site 2 is located on the municipalities of Moëze and Saint-Froult.

(Statzner *et al*. [34]), and (ii) stronger in carabids compared to spiders [27,35] because of the higher phylogenetic proximity of species with conserved and functionally adapted traits in salt-marsh carabids [36,37].

Hypothesis 2: despite the fact that TD influences the strength of the correlation between PD and FD by side effects [9,38,39], we expect a relationship between TD and both PD and FD stronger for carabids owing to the greater sensitivity of carabids to environmental constraints such as salinity [40], resulting in a pool of species with closely functional traits for e.g. resisting salinity, avoiding flooding and/or recolonizing the marsh after tides. The other way around, we expect spider assemblages to be more driven by changes in vegetation structure which has been shown in coastal [41] as well as in other inland (e.g. [42]) habitats.

Hypothesis 3: environmental stressors are expected to influence TD in the same direction for each taxon, with e.g. salinity having a negative influence on TD of both spiders and carabids [40,43,44]. In addition, the environmental factors should influence the PD of salt-marsh organisms similarly because of their recent (less than 6000 years: [45]) evolutionary history in salt marshes. By contrast, environmental variables affecting FD are expected to differ between taxa, as reported for other ecosystems [46].

# 2. Material and methods

## 2.1. Study sites and sampling design

The study was conducted on two salt marshes in Charente-Maritime (New Aquitaine Region, France). The first site (site 1) is located on the Ile de Ré, and the second site (site 2) is located on the municipalities of Moëze and Saint-Froult (figure 1*a*). This study focuses on the parts of each site that are classified as national nature

reserve, and selected because they have a large surface area of coastal salt marshes (EUNIS A2.5, Coastal saltmarshes and saline reedbeds, according to the European Nature Information System habitats typology). These habitats that are considered to be of high heritage value were targeted in this study.

Three zones per site were defined according to their topographic and vegetation characteristics (successively: high, middle and low marsh), using EUNIS typology. These habitats are representative of different immersion frequencies and resulting salinity gradient at each site. Therefore, we identified zones within which immersion frequencies and vegetation are considered homogeneous. On site 1, the zones were distant and fragmented, a consequence of past salt farming (figure 1b). High marshes have about 0.5% of annual recovering tides whereas low marshes have about 60% (based on field observations by the managers of both reserves). On site 2, the zones were continuous from the beginning to the upper salt marshes (figure 1c). The frequency of tides recovering the marsh decreases along a land-sea transect, ranging from 10% of annual tides in the high marsh to 35% in the low marsh.

Pitfall traps, consisting of cylindrical PVC pipes (diameter of 9 cm) that are buried on the ground so that the upper edges of the pipe and the ground are on the same level, were used to sample ground-active spiders and carabids [47]. Inside the pipe, a plastic jar was inserted. Over the plastic jar, a funnel attached to the outer edges of the PVC tube was placed. Finally, the roof, supported by metal stakes, was placed over the device to prevent the collecting liquid from becoming diluted with rain.

The collecting plastic jar was filled with three-quarters of a 250 g $l^{-1}$ saline solution supplemented with a drop of dishwashing liquid per litre (to reduce the surface tension of the liquid). Each of the study sites had three zones, each with a total of three sampling stations to provide sufficient spatial replication. Each station included four pitfall traps arranged linearly. The pitfall traps, with an interception radius of approximately 5 m (see [48] and [49] for spiders and carabids, respectively), were placed at the centre of non-overlapping circles of 10 m diameter, in order to prevent them from overlapping each other. A total of 36 traps were set up per study site. Five trapping sessions of 4–12 days each were carried out between April and July 2019. The dates and duration of the sampling sessions differed between sites, along with differences in immersion frequencies. All sessions accumulated, 29 days of sampling were done on the first site, and 48 days on the second site. All adult carabids and spiders were identified down to species level except for individuals of *Dyschirius* (Carabidae) that were grouped into spp.

## 2.2. Environmental variables

When setting the pitfall traps, environmental variables were recorded within a 5 m radius circle around each pitfall trap (matching the theoretical interception areas of traps), and were then averaged for each zone. Litter depth and average and maximum heights of vegetation were measured to the nearest centimetre using a decimetre and a metre, respectively (table 1a). The percentage of bare ground was estimated visually, and the soil salinity was estimated by measuring the conductivity of 1 g of soil diluted in 15 ml of distilled water, then converted into a mass of $NaCl\,l^{-1}$. Within the same area, a phytosociological survey [50] was carried out by estimating the recovery percentage to all plant species present, according to Braun-Blanquet scale [51] in order to verify that zones were homogeneous in terms of vegetation (table 1b).

## 2.3. Phylogenetic tree building

Phylogenetic trees were constructed by combining phylogenetic and taxonomic data from the literature, with the assumption that the identical branch lengths between genera and species were 1 and 0.5, respectively, as distances were not available for all species (also, we wanted to be able to compare results between taxa). The phylogenetic tree of spiders (appendix A, figure 4) was adapted from that of Wheeler *et al.* [52] by adding missing genera using Arnedo *et al.* [53], Frick *et al.* [54], and Wang *et al.* [55] for Linyphiidae & Millidge [56], Agnarsson [57], Maddison [58], Azevedo *et al.* [59], Piacentini & Ramírez [60] and Scharff *et al.* [61] for other families. The phylogenetic tree of carabids (appendix B, figure 5) was adapted from that of Martínez-Navarro, Galián, & Serrano [62], Sasakawa & Kubota [63], Ober & Maddison [64], Ruiz Jordal & Serrano [65] and López-López & Vogler [66].

## 2.4. Functional traits used

In order to calculate the FD per pitfall trap for each taxon, functional traits were assigned to each of the spider and carabid species according to the literature cited in appendix C, table 3. The selected traits, *viz.* size, dispersal capacity and overall diet, were chosen to be (i) relevant traits for computing FD of each

**Table 1.** Environmental variables (mean ± s.d., $n = 12$) for each salt-marsh zone and for each site. (Successive letters indicate significant differences by ANOVA test followed by Tukey post-hoc tests or Kruskal–Wallis or Mann–Whitney or Welch tests, where appropriate. Bonferroni correction was used for post-hoc tests if necessary. Plant species are given only when they occur in more than 75% of the surveys carried out in the area and had a Braun-Blanquet coverage > 1.5. HM, high marsh; MM, middle marsh; LM, low marsh.)

| | | | site 1 | | site 2 | |
|---|---|---|---|---|---|---|
| (*a*) | litter depth (cm) | HM | 0.3 ± 0.2 | a | 0.2 ± 0.3 | A |
| | | MM | 1.0 ± 0 | b | 0.1 ± 0.1 | A |
| | | LM | 0 ± 0 | c | 0.1 ± 0 | A |
| | average height of vegetation (cm) | HM | 48.8 ± 7.1 | a | 50.0 ± 13.6 | A |
| | | MM | 22.0 ± 5.4 | b | 39.2 ± 4.2 | B |
| | | LM | 21.5 ± 3.4 | b | 33.7 ± 3.1 | C |
| | maximum height of vegetation (cm) | HM | 95.0 ± 11.7 | a | 95.8 ± 10.8 | A |
| | | MM | 41.3 ± 14.8 | b | 70.0 ± 18.1 | B |
| | | LM | 78.3 ± 24.1 | a | 59.2 ± 6.7 | B |
| | percentage of bare soil (%) | HM | 3.3 ± 2.3 | a | 1.1 ± 0.7 | A |
| | | MM | 15.6 ± 9.5 | b | 0.5 ± 0.1 | B |
| | | LM | 4.4 ± 4.0 | a | 0.9 ± 0.2 | A |
| | soil salinity (NaCl: g kg$^{-1}$) | HM | 14.0 ± 9.9 | a | 13.1 ± 6.6 | A |
| | | MM | 16.8 ± 3.2 | b | 4.3 ± 2.6 | B |
| | | LM | 24.1 ± 5.3 | b | 13.5 ± 6.2 | A |
| (*b*) | dominant plant species (mean of Braun-Blanquet scale values) | HM | *Halimione portulacoides* 2.8 ± 0.8 | | *Halimione portulacoides* 2.7 ± 0.75 | |
| | | | *Arthrocnemum fruticosum* 2.5 ± 0.5 | | *Arthrocnemum fruticosum* 4.1 ± 0.9 | |
| | | | *Inula crithmoides* 1.8 ± 0.6 | | | |
| | | MM | *Halimione portulacoides* 3.2 ± 0.6 | | *Halimione portulacoides* 5.0 ± 0 | |
| | | | *Arthrocnemum perenne* 2.0 ± 0 | | *Arthrocnemum fruticosum* 0.9 ± 0.5 | |
| | | LM | *Halimione portulacoides* 3.2 ± 0.4 | | *Halimione portulacoides* 2.0 ± 0.8 | |
| | | | *Arthrocnemum perenne* 2.4 ± 0.5 | | *Arthrocnemum perenne* 1.7 ± 0.4 | |
| | | | *Spartina maritima* 1.6 ± 0.5 | | *Puccinellia maritima* 4.1 ± 0.7 | |

group, and (ii) comparable between the two taxa. Long-distance dispersal, body size and overall diet were chosen to broadly reflect species colonization ability, stress tolerance capacities and diversity of prey resources, respectively (e.g. [67]). The risk of considering only functional traits related to salt-marsh stressors is to study assemblages homogenized by the dominance of salt-marsh adapted species, i.e. high abundance of small, aeronaut, halophilic species [40].

## 2.5. Statistical analysis

TD was estimated by measuring the species richness of samples per trap (both sites together), and computed using the BAT package [68] following the methods described by Lopez *et al*. [69] for

**Table 2.** Initial and final SEM models for spiders and carabid beetles. (All models included site/zone as a random factor and correlated error between PD and FD. PD, phylogenetic diversity; FD, functional diversity; TD, taxonomic diversity.)

| | | |
|---|---|---|
| spiders | initial model | PD ~ salinity + vegetation height + TD |
| | | FD ~ PD + litter depth + TD |
| | | TD ~ vegetation height |
| | final model | PD ~ salinity + TD |
| | | FD ~ PD + litter depth + TD |
| carabids | initial model | PD ~ salinity + vegetation height + TD |
| | | FD ~ PD + vegetation height + TD |
| | | PD ~ salinity + vegetation height |
| | final model | PD ~ TD |
| | | FD ~ PD + TD |
| | | TD ~ salinity |

obtaining corrected jackknife estimators. Final TD was calculated by averaging the corrected jackknife estimates in order to account for sampling variability, and checked with the accuracy function (scaled mean squared error <5% in each case). In the same manner, PD and FD were estimated using the Petchey & Gaston [4,5] estimator for FD and Faith [70] for PD, following the methods described by Cardoso *et al.* [71] using the BAT package. Final PD and FD were also calculated by averaging the corrected jackknife estimates. Distance matrix for phylogenetic distances was calculated using Gower distance from the FD package [72].

The correlation between PD and FD was estimated in a Bayesian framework with a Student's *t*-distribution (which reduces sensitivity to outliers) using the brms package [73]. We used 2000 iterations on four chains. Model convergence was checked by visually inspecting diagnostic plots.

To select environmental variables affecting PD and FD (for later use in structural equation models (SEM)), models were built within a Bayesian framework using brms [73] with two chains and default priors. All environmental variables were standardized and centred. The models included salinity, bare ground, litter depth, mean vegetation height, maximum vegetation height, and site and salt-marsh zone as a random factor. Model convergence was checked by visually inspecting diagnostic plots and using the Rhat value. Parameter selection was based on 'HDI + ROPE decision rule' [74] with a determined range value of between −0.1 * s.d.($y$) and 0.1 * s.d.($y$) [74], and was performed using bayestestR [75]. This rule states that if the HDI is completely outside the ROPE, one can reject the 'null hypothesis' for the particular parameter. Symmetrically, if the ROPE completely covers the HDI, one can accept the null hypothesis. Otherwise, whether to accept or reject the null hypothesis remains undecided. Variables were selected as candidates for the SEM when ROPE > 95%, which means we accepted variables for which we could reject the null hypothesis and variables for which we could not decide whether or not to reject the null hypothesis (under the limit of 5%). We also provided the probability of direction (pd), which is the probability that the posterior distribution of a parameter is strictly positive or negative.

We assessed the relative contribution of environmental variables selected by Bayesian models using SEM. The SEM approach was used to assess the indirect effect of TD on PD and FD as their calculations both depend on TD, and to test for correlated errors between PD and FD. A significant correlated error between the two variables indicates the existence of an unknown parameter influencing both variables. We used the piecewise SEM package [76] as it allows us to use mixed models in association with the nlme package [77]. Our initial model included the following links: (i) PD is affected by TD and selected environmental variables; (ii) FD is affected by TD, selected environmental variables, and PD; (iii) TD is affected by selected environmental variables; and (iv) there is a correlated error between PD and FD (table 2). Site was used as a random factor in every link model using nlme [77]. After the specification of the initial model, we re-defined our model excluding non-significant links ($p < 0.05$) using a stepwise approach until ΔAICc < 2 between two subsequent models. Finally, we assessed model fit using Fisher's C statistic. All statistical analyses were performed using R STUDIO software (v. 3.5.1).

# 3. Results

## 3.1. Habitat characteristics

Litter depth and vegetation height (average and maximum) tended to decrease from high to low marsh, with some differences noted for site 1 (table 1a). The percentage of bare ground was maximum in the middle marsh of site 1, and minimal in the same area of site 2. Salinity was globally increasing from high to low marsh, although this variable was minimal in the middle marsh of site 2. According to a phytosociological survey, low marshes were characterized by the dominance of *Halimione portulacoides* in site 1 as against the dominance of *Puccinellia maritima* in site 2 (table 1b). The middle marshes of both sites were dominated by *H. portulacoides*. Finally, for high marshes a co-dominance of *H. portulacoides* and *Arthrocnemum fruticosum* was found in site 1; by contrast, a dominance of *A. fruticosum* only was found in site 2.

## 3.2. Description of assemblages

A total of 3359 adult spiders belonging to 55 species, of which 58.9% of individuals sampled are considered halophilic (appendix D, table 4), were collected by pitfall traps [78]. Spiders had an average size of 7.13 ± 3.79 mm. Hunting guilds were dominated by ground-hunting individuals (56.6%), and for dispersal methods, most individuals were ballooners (63.4%). A total of 4005 carabids belonging to 12 species, of which 99.7% of individuals sampled are considered halophilic (appendix E, table 5), were collected by pitfall traps. Carabids had an average size of 6.82 ± 0.84 mm. The diet of carabids was dominated by generalist predator individuals (99.3%), and the main dispersal technique was represented by polymorph individuals (92.5%).

## 3.3. Taxonomic, phylogenetic and functional indices

A table containing all values can be found in appendix F, table 6.

## 3.4. Correlations between phylogenetic and functional diversities

The correlation factors between PD and FD were 0.48 (95% confidence interval (CI): 0.27–0.66) and 0.89 (95% CI: 0.83–0.94) for spiders and carabids, respectively (figure 2).

## 3.5. Environmental variable selection

The Bayesian model for spider PD successfully converged and had an $R^2 = 0.271$. Mean vegetation height and salinity were the best explanatory variables. Mean vegetation height effect on spider PD had a high probability of existing (pd = 98.65%, median = 8.49, 89% CI (2.70, 14.37)), and could be considered significant (0% in ROPE). Salinity effect on spider PD had a high probability of existing (pd = 99.7%, median = 7.14, 89% CI (3.57, 11.11)), and could be considered as significant (0% in ROPE). The model for spider FD successfully converged and had an $R^2 = 0.415$. Litter depth was the best explanatory variable. Litter depth effect on spider FD had a high probability of existing (pd = 99.6%, median = −0.86, 89% CI (−1.31, −0.35)), and could be considered significant (0% in ROPE). The model for spider TD successfully converged and had an $R^2 = 0.442$. Mean vegetation height was the best explanatory variable. Mean vegetation height effect on spider TD had a high probability of existing (pd = 99.9%, median = 4.76, 89% CI (2.21, 7.29)), and could be considered significant (0% in ROPE).

The Bayesian model for carabid PD successfully converged and had an $R^2 = 0.188$. Mean vegetation height and salinity were the best explanatory variables. Mean vegetation height effect on carabid PD had a medium probability of existing (pd = 97.75%, median = −0.58, 89% CI (−0.99, −0.11)), and its significance remained undecided (7.58% in ROPE). Salinity effect on carabid PD had a medium probability of existing (pd = 95.75%, median = −1.79, 89% CI (−3.33, 0.31)), and its significance remained undecided (7.00% in ROPE). The model for carabid FD successfully converged and had an $R^2 = 0.18$. Mean vegetation height was the best explanatory variable. Mean vegetation height effect on carabid FD had a high probability of existing (pd = 99.6%, median = −0.86, 89% CI (−1.31, −0.35)), but its significance remained undecided. The model for carabid TD successfully converged and had an $R^2 = 0.222$. Mean vegetation height and salinity were the best explanatory variables. Mean vegetation height effect on carabid TD had a high probability of existing (pd = 99.15%, median = −1.81, 89% CI (−2.92, −0.71)), and could be considered significant (0% in ROPE).

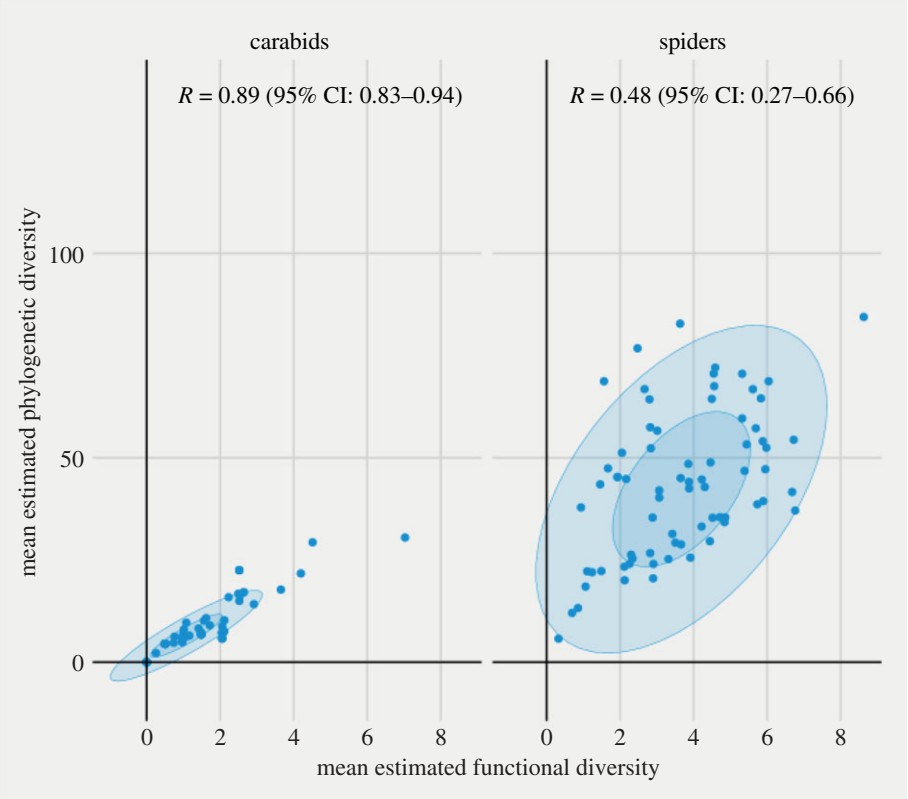

**Figure 2.** Plot of mean phylogenetic diversity as a function of functional diversity for spiders and carabids. Ellipses correspond to 5% and 95% confidence intervals.

Salinity effect on carabid TD had a probability of existing (pd = 97.70%, median = −0.89, 89% CI (−1.552, −0.219)), still its significance remained undecided (2.9% in ROPE).

## 3.6. Effects of taxonomic diversity and environmental variables on phylogenetic diversity-functional diversity relationships

When testing the relationships between the different diversity metrics and environmental variables for spiders, our final SEMs indicated good fit with the data (Fisher's $C = 1.075$, $p = 0.898$; figure 3).

Salinity and litter depth were linked to spider PD (positive link, selected in the model but only marginally significant: $p = 0.068$) and FD (negative link, selected in the model and again almost significant: $p = 0.051$), respectively. Spider PD was strongly and positively related to TD (coefficient standard estimate: 0.705). Spider FD was positively linked to PD (coefficient standard estimate: 0.231) and TD (coefficient standard estimate: 0.435).

When testing the relationships between the different diversity metrics and environmental variables for carabids, our final SEMs indicated good fit with the data (Fisher's $C = 5.368$, $p = 0.252$; figure 3). Salinity was negatively linked to carabid TD (link, selected in the model but not significant: $p = 0.110$). Carabid PD was strongly and positively related to TD (coefficient standard estimate: 0.820). Carabid FD was positively linked to PD (coefficient standard estimate: 0.452) and TD (coefficient standard estimate: 0.520). Correlated errors were found between FD and PD (coefficient standard estimate: 0.497).

# 4. Discussion

## 4.1. Correlations between phylogenetic and functional diversities

As expected, positive correlations between PD and FD diversity metrics were found for both spiders and carabids. This was probably owing to the fact that functionally adapted species are also often

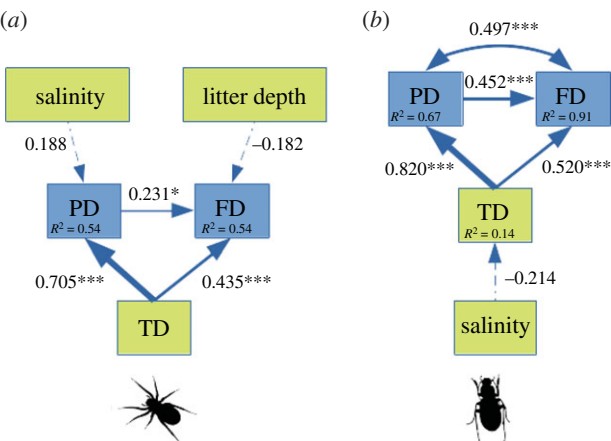

**Figure 3.** Best piecewise SEMs showing links between taxonomic (TD), phylogenetic (PD) and functional (FD) diversities and environmental variables for: (*a*) spiders and (*b*) carabids. Thickness of arrows is proportional to the standardized path coefficients (directionality and size given within boxes). Asterisks indicate the significance level of linkages (<0.1, *<0.05, **<0.01, ***<0.001), and the dashed lines correspond to paths included but not significant (*p* > 0.05). Double arrows represent correlated errors. Conditional $R^2$ values are given within the boxes containing variables.

phylogenetically close [11]. Interestingly, this correlation between PD and FD is expected to increase with the number of functional traits used [79]; however, only three functional traits were used to homogenize the number and nature between the two studied taxa. This relationship is strong, even with a small number of traits used, which suggests that functional traits of species in salt marshes are phylogenetically conserved. Interestingly, this hypothesis, which was already stressed by Cadotte *et al.* [12], requires further studies. In addition, the relatively small number of species collected in the studied salt marshes (*n* = 55 for spiders and *n* = 12 for carabids) compared to that in less constrained environments from the same biogeographic area (e.g. see [80]: *n* = 99 for spiders and *n* = 43 for carabids) may also increase the strength of the PD–FD correlation [10], especially for carabids.

The observed correlation between PD and FD can also result from the parameters of phylogenetic trees used. There is evidence that the strength of the PD–FD correlation can be increased with the use of phylogenetic trees that are symmetrical and/or have long terminal branches [10]. Here, it was difficult to calculate an index of symmetry and/or length of the terminal branches because the branches had similar lengths between genus and species owing to limited knowledge in the phylogeny of arthropods. Finally, it is also possible that the inclusion of TD in the calculation of PD and FD influences the relationship between these two metrics ([9,38]: see discussion below).

The correlation between PD and FD was stronger for carabids than for spiders, corroborating our initial hypothesis. First, the reduced number of salt-marsh carabids compared to that of salt-marsh spiders (12 species for carabids as against 55 species for spiders) increased this correlation [10]. Moreover, carabids that adapted to salt marshes are phylogenetically closer than spiders that adapted to salt marshes owing to a more recent evolutionary radiation of carabids in this ecosystem (appendix B, figure 5; see [36,37]). In fact, it should be noted that the halophilic species of spiders are not concentrated in the same genus ([27,35], see also appendix A, figure 4), but rather result from adaptations, independent of their phylogeny (trait convergence: [81]). Therefore, the observed relationship between PD and FD suggests that functional traits are phylogenetically more conserved in carabids than in spiders.

## 4.2. Effects of taxonomic diversity on phylogenetic diversity-functional diversity relationships

First, the SEMs carried out on both taxa revealed that the links between PD and FD were weaker than the links between TD and both PD and FD (figure 3). These results suggest that the relationship between PD and FD is mainly affected by a side effect owing to the inclusion of TD in both PD and FD calculation, which corroborates with previous studies [9,38,39]. This correlation is usually increasing with a decreasing number of species [10], which could also explain the pattern we observed in carabids (low TD, high $R^2$) versus spiders (higher TD, lower $R^2$).

Expectedly, TD was also found to be more strongly related to FD and PD in carabids than in spiders. Because of their ecology (larval stage in the soil), carabids are more exposed to salinity than spiders and

possess particular morphological adaptive features such as a waterproof-like inter-tegument cuticle [43]. This results in a highly specialized halophilic pool for this taxon (99.7% of individuals are halophilic, in opposition to 58.9% of halophilic individuals for spiders). Furthermore, as stated above, halophilic carabids seem to be more phylogenetically clustered than halophilic spiders (appendices A and B, figures 4 and 5), resulting in a strong link between the TD and both PD and FD for the taxa. The lower percentage of halophilic individuals (58.9%) compared to that of halophilic carabid individuals (99.7%) obtained in this study is consistent with this hypothesis.

## 4.3. Effects of environmental filtering on diversity metrics

Remarkably, salinity influenced the TD of carabids because of their greater sensitivity to this stressor (see above and [40,82])—but did not influence the TD of spiders. The importance of this factor in structuring spider assemblages has been reported in previous field studies [40,83]. This unexpected result could be explained by the fact that spiders are more plastic to saline stress than carabids [27,40], and thus, more diverse non-specialized spiders could live in salt-marsh habitats. Laboratory experiments have revealed repeatedly that halophilic spiders, although strictly restricted to salt marshes, do perform better (in terms of both survival and fitness) without saline stress than under saline to hyper-saline stress [84,85]. Salinity probably had a strong influence on carabids because of their greater sensitivity to this stress (see above and [40,82]).

Interestingly, PD was differentially influenced by the salinity for spiders and by an unidentified variable for carabids. These results are consistent with the hypothesis that spiders have higher plasticity in response to salinity [27,40]. This higher plasticity can result in a less specialized pool of spider species as seen from the low percentage of halophilic species recorded in traps. Because environmental filters act more strongly on carabids, it seems logical that salinity had little influence on their PD because it is composed of a pool of species already strongly selected based on the factors.

The main environmental variable driving the FD of carabids and spiders differed between these taxa: influence of litter depth for spiders and unidentified variable for carabids, which corroborates our initial expectation. These results indeed support the idea that the drivers acting on the FD of these taxa are different [46,86–88]. The effect of litter depth on spider FD can be explained by a change in hunting guilds driven by a modification of prey composition with litter depth (see also [89,90]). It is important to point out that the unidentified environmental variable affecting the FD of carabids is the same as the one driving their PD (as indicated by the strong correlated error found between the two response variables), thus strengthening the links between these two metrics. Parameters, not measured in this study, such as soil moisture and the duration of flooding by the adjacent sea, are known to affect the FD of carabids [46], and can certainly be a candidate for this unidentified variable. Sediment granulometry is also known as a filter of carabid FD that drives the endogenous larval life and the full-grown burrowing-type strategy of carabids [91].

The SEM also highlighted that salt-marsh zonation, based on vegetation assemblages, did not influence neither PD nor FD, suggesting that the driving environmental variables are similar in both study sites. Environmental filters rather act at a landscape scale for both taxa studied (see also [31,92]). Therefore, it is also possible that for carabids, the variable influencing both PD and FD acts at a landscape scale.

In conclusion, both spiders and carabids exhibited high correlations between FD and PD, reinforcing the importance of considering these metrics simultaneously in conservation studies [3,6]. Interestingly, the environmental factors driving FD and PD differed between taxa, and this, together with the percentage of specialist species that also differed between the two groups, suggest that these two dominant groups of ground-dwelling arthropods differentially react to stressful factors. Further studies should investigate the role of other factors, both local (e.g. soil texture) and landscape (e.g. spatial heterogeneity), driving diversity metrics of predatory arthropods in salt marshes. Finally, our study highlights that even in taxa of the same phylum and occupying the same niche in a highly constrained habitat, FD and PD can have different drivers, showing different filtering mechanisms and evolutionary history at small spatial and temporal scales.

Data accessibility. Data and Rcode are available on the Dryad Digital Repository (https://doi.org/10.5061/dryad. dfn2z3506) [78].
Authors' contributions. Conception and design: J.P. Acquisition of data: A.R. and P.D. Analysis and interpretation: A.R. and D.L. Drafting: A.R., D.L. Revising the article: J.P. and T.L.-L.
Competing interests. We declare we have no competing interests.

Funding. This work is a contribution to the project PAMPAS ANR-18-CE32-0006, supported by the French National Research Agency. J.P. and D.L. are supported by the Agence de l'Eau Loire-Bretagne and the Région Bretagne (PEPPS project).

Acknowledgements. We would like to thank the Editage company for English editing, one anonymous referee for providing relevant comments and suggestions, and the nature reserve staff of Möeze-Oléron and Lilleau des Niges for field assistance: Fréderic Robin, Jean-Christophe Lemesle, Philippe Delaporte, Vincent Lelong, Julien Gernigon, and Lucas Deplaine.

# Appendix A

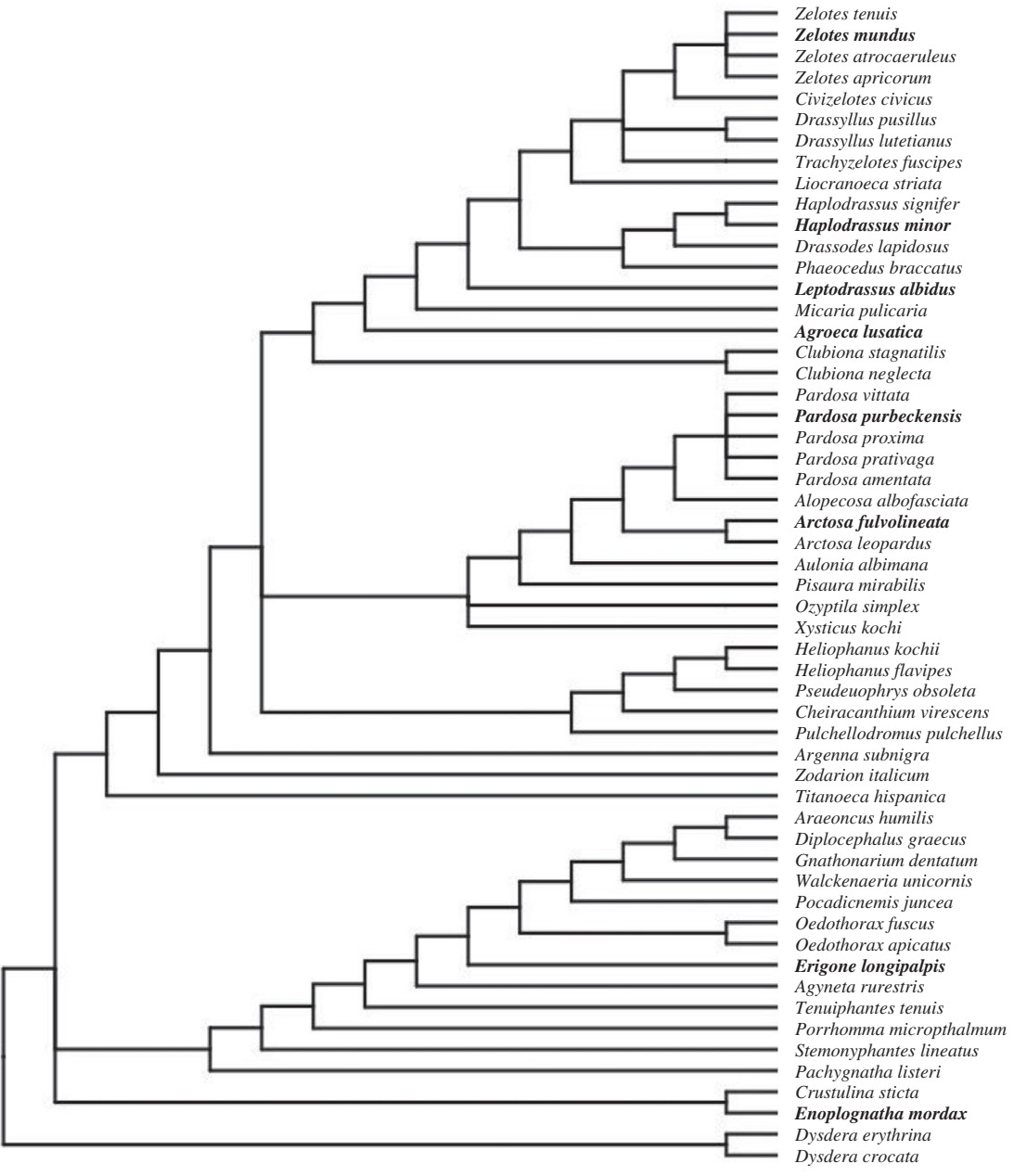

**Figure 4.** Phylogenetic tree of sampled spider species. (Halophilic species are in bold according to Pétillon *et al.* [100].)

# Appendix B

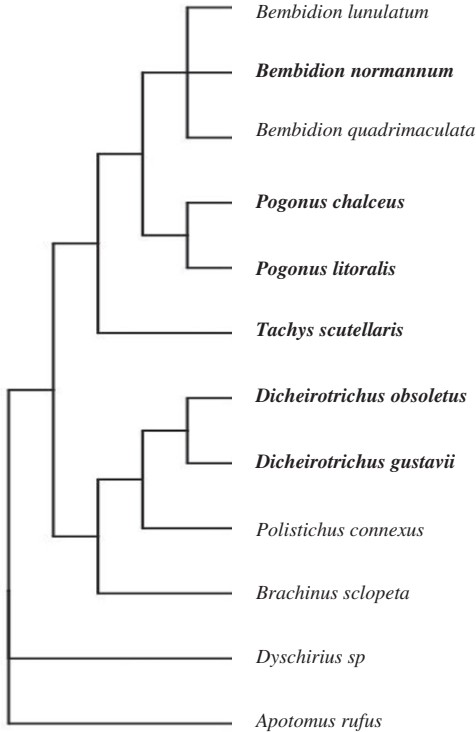

**Figure 5.** Phylogenetic tree of sampled carabid species. (Halophilic species are in bold according to Pétillon *et al*. [100] and Georges *et al*. [101].)

# Appendix C

**Table 3.** Functional traits used for spiders and carabids. (Attributes of traits were selected according to: Bonte *et al*. [93], Lambeets *et al*. [94,95], Albrecht *et al*. [96], Schirmel *et al*. [86], Fischer *et al*. [97], Schirmel & Buchholz [87], Schirmel *et al*. [88], Gobbi *et al*. [98] and Torma *et al*. [99].)

| spiders | | carabids | |
|---|---|---|---|
| traits | attribute values | traits | attribute values |
| maximum body size of females | continuous trait | maximum body size | continuous trait |
| ballooning | yes-no | flight capacity | brachypterous; dimorphic; macropterous; polymorphic |
| hunting strategy | ground hunter; vegetation hunter; ambush hunter; running hunter; wandering web weaver; sheet weaver; spaced web weaver; orbital web weaver | type of diet | omnivorous; phytophagous; predators |

# Appendix D

**Table 4.** List of spider species sampled at site 1 [1] and site 2 [2], and functional traits of species. (The halophilic species are annotated with an asterisk, and considered halophilic according to Pétillon *et al*. [100], the others are considered non-halophilic.)

| species sampled | site | maximum body size of females (mm) | hunting strategy | ballooning |
|---|---|---|---|---|
| *Agroeca lusatica* (L. Koch, 1875)* | [2] | 7 | ground hunter | no |
| *Agyneta rurestris* (C. L. Koch, 1836) | [1, 2] | 2.4 | wandering web weaver | yes |
| *Alopecosa albofasciata* (Brullé, 1832) | [1] | 12.0 | ground hunter | no |
| *Araeoncus humilis* (Blackwall, 1841) | [2] | 1.8 | wandering web weaver | yes |
| *Arctosa fulvolineata* (Lucas, 1846)* | [1, 2] | 12.0 | ground hunter | no |
| *Arctosa leopardus* (Sundevall, 1833) | [2] | 9.5 | ground hunter | no |
| *Argenna subnigra* (O. Pickard-Cambridge, 1861) | [1, 2] | 2.5 | ground hunter | no |
| *Aulonia albimana* (Walckenaer, 1805) | [1] | 4.5 | ground hunter | yes |
| *Cheiracanthium virescens* (Sundevall, 1833) | [2] | 7 | vegetation hunter | no |
| *Civizelotes civicus* (Simon, 1878) | [2] | 5.4 | ground hunter | no |
| *Clubiona neglecta* (O. Pickard-Cambridge, 1862) | [2] | 8 | vegetation hunter | no |
| *Clubiona stagnatilis* (Kulczyński, 1897) | [2] | 8 | vegetation hunter | yes |
| *Crustulina sticta* (O. Pickard-Cambridge, 1861) | [2] | 2.5 | spaced web weaver | no |
| *Diplocephalus graecus* (O. Pickard-Cambridge, 1873) | [1] | 2.2 | wandering web weaver | no |
| *Drassodes lapidosus* (Walckenaer, 1802) | [1] | 18.0 | ground hunter | no |
| *Drassyllus lutetianus* (L. Koch, 1866) | [1, 2] | 7.5 | ground hunter | yes |
| *Drassyllus pusillus* (C. L. Koch, 1833) | [1, 2] | 5.0 | ground hunter | yes |
| *Dysdera crocata* (C. L. Koch, 1838) | [1] | 15.0 | ground hunter | no |
| *Dysdera erythrina* (Walckenaer, 1802) | [1] | 10.0 | ground hunter | no |
| *Enoplognatha mordax* (Thorell, 1875)* | [1, 2] | 8.5 | spaced web weaver | yes |
| *Erigone longipalpis* (Sundevall, 1830)* | [1, 2] | 3.5 | wandering web weaver | yes |
| *Gnathonarium dentatum* (Wider, 1834) | [2] | 3 | wandering web weaver | yes |
| *Haplodrassus minor* (O. Pickard-Cambridge, 1879)* | [2] | 4 | ground hunter | no |
| *Haplodrassus signifer* (C. L. Koch, 1839) | [1] | 9.0 | ground hunter | no |
| *Heliophanus flavipes* (Hahn, 1832) | [1] | 6.0 | running hunter | no |
| *Heliophanus kochii* (Simon, 1868) | [1] | 5.3 | running hunter | no |
| *Leptodrassus albidus* (Simon, 1914)* | [1, 2] | 1.5 | ground hunter | no |
| *Liocranoeca striata* (Kulczyński, 1882) | [1] | 5.5 | ground hunter | no |
| *Micaria pulicaria* (Sundevall, 1831) | [2] | 4.5 | ground hunter | no |
| *Oedothorax apicatus* (Blackwall, 1850) | [2] | 3.3 | wandering web weaver | yes |
| *Oedothorax fuscus* (Blackwall, 1834) | [2] | 2.9 | wandering web weaver | yes |

(*Continued.*)

| species sampled | site | maximum body size of females (mm) | hunting strategy | ballooning |
|---|---|---|---|---|
| *Ozyptila simplex* (O. Pickard-Cambridge, 1862) | [2] | 5 | ambush hunter | yes |
| *Pachygnatha listeri* (Sundevall, 1830)* | [2] | 5 | orbital web weaver | yes |
| *Pardosa amentata* (Clerck, 1757) | [1, 2] | 8.0 | ground hunter | yes |
| *Pardosa prativaga* (L. Koch, 1870) | [2] | 6 | ground hunter | yes |
| *Pardosa proxima* (C. L. Koch, 1847) | [2] | 6.5 | ground hunter | yes |
| *Pardosa purbeckensis* (F. O. Pickard-Cambridge, 1895)* | [2] | 9.0 | ground hunter | yes |
| *Pardosa vittata* (Keyserling, 1863) | [2] | 6.3 | ground hunter | yes |
| *Phaeocedus braccatus* (L. Koch, 1866) | [1] | 6.5 | ground hunter | no |
| *Pisaura mirabilis* (Clerck, 1757) | [2] | 15 | ambush hunter | yes |
| *Pocadicnemis juncea* (Locket & Millidge, 1953) | [1, 2] | 2.2 | wandering web weaver | no |
| *Porrhomma microphtalmum* (O. Pickard-Cambridge, 1871) | [1, 2] | 2.2 | sheet weaver | yes |
| *Pseudeuophrys obsoleta* (Simon, 1868)* | [1] | 4.8 | running hunter | no |
| *Pulchellodromus pulchellus* (Lucas, 1846) | [1, 2] | 4.6 | ambush hunter | no |
| *Stemonyphantes lineatus* (Linnaeus, 1758) | [1, 2] | 6.4 | wandering web weaver | no |
| *Tenuiphantes tenuis* (Blackwall, 1852) | [1, 2] | 3.2 | sheet weaver | yes |
| *Titanoeca hispanica* (Wunderlich, 1995) | [1, 2] | 4.8 | spaced web weaver | no |
| *Trachyzelotes fuscipes* (L. Koch, 1866) | [1, 2] | 6.0 | ground hunter | no |
| *Walckenaeria unicornis* (O. Pickard-Cambridge, 1861) | [1, 2] | 3.1 | wandering web weaver | no |
| *Xysticus kochi* (Thorell, 1872) | [1, 2] | 8.0 | ambush hunter | yes |
| *Zelotes apricorum* (L. Koch, 1876) | [1] | 9.0 | ground hunter | no |
| *Zelotes atrocaeruleus* (Simon, 1878) | [1] | 8.1 | ground hunter | no |
| *Zelotes mundus* (Kulczyński, 1897)* | [2] | n.a. | ground hunter | no |
| *Zelotes tenuis* (L. Koch, 1866) | [1, 2] | 9.6 | ground hunter | no |
| *Zodarion italicum* (Canestrini, 1868) | [1] | 3.0 | ambush hunter | no |

# Appendix E

**Table 5.** List of carabid species sampled at site 1 [1] and site 2 [2], and functional traits of species. (The halophilic species are annotated with an asterisk, and considered halophilic according to Pétillon *et al*. [100] and Georges *et al*. [101], the others are considered non-halophilic.)

| species sampled | site | maximum body size (mm) | flight capacity | type of diet |
|---|---|---|---|---|
| *Apotomus rufus* (Rossi, 1790) | [2] | 4.5 | macropterous | omnivorous |
| *Bembidion lunulatum* (Geoffroy, 1785) | [1] | 3.5 | macropterous | omnivorous |
| *Bembidion normannum* (Dejean, 1831)* | [1] | 3.5 | macropterous | omnivorous |
| *Bembidion quadrimaculata* (Linnaeus, 1760) | [1, 2] | 3.2 | macropterous | omnivorous |
| *Brachinus sclopeta* (Fabricius, 1792) | [2] | 7.0 | macropterous | omnivorous |
| *Dicheirotrichus gustavii* (Crotch, 1871)* | [1, 2] | 7.5 | macropterous | phytophagous |
| *Dicheirotrichus obsoletus* (Dejean, 1829)* | [1, 2] | 8 | macropterous | phytophagous |
| *Dyschirius* spp. | [1] | n.a. | macropterous | omnivorous |
| *Pogonus chalceus* (Marsham, 1802)* | [1, 2] | 7 | polymorphic | omnivorous |
| *Pogonus litoralis* (Duftschmid, 1812)* | [1, 2] | 8 | macropterous | omnivorous |
| *Polistichus connexus* (Geoffroy in Fourcroy, 1785) | [1, 2] | 9.5 | macropterous | omnivorous |
| *Tachys scutellaris* (Stephens, 1828)* | [1, 2] | 2.6 | macropterous | omnivorous |

# Appendix F

**Table 6.** Values of taxonomic (TD) functional (FD) and phylogenetic (FD) diversities, for each site (FA = fier d'Ars, MO = Moëze Brouage) and for each taxon with associated standard deviation.

| site | FA | MO |
|---|---|---|
| TD spiders | 11.52 ± 8.84 | 19.56 ± 6.25 |
| FD spiders | 2.90 ± 1.84 | 4.43 ± 1.3 |
| PD spiders | 44.64 ± 20.72 | 42.48 ± 14.81 |
| TD carabids | 3.08 ± 3.14 | 3.77 ± 2.86 |
| FD carabids | 1.34 ± 1.42 | 1.46 ± 0.96 |
| PD carabids | 7.68 ± 8.03 | 8.69 ± 4.96 |

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
