## [Peer Review File · Royal Society Open Science]

Review History

RSOS-202093.R0 (Original submission)

Review form: Reviewer 1

Is the manuscript scientifically sound in its present form?

No

Are the interpretations and conclusions justified by the results?

No

Is the language acceptable?

Yes

Do you have any ethical concerns with this paper?

No

Have you any concerns about statistical analyses in this paper?

No

Recommendation?

Major revision is needed (please make suggestions in comments)

Comments to the Author(s)

This study examines the relationship between functional and phylogenetic diversity of two dominant ground dwelling arthropods in saltmarshes and the effect of environmental variables on them considering the influence of taxonomic diversity. This is a very interesting topic, as biodiversity has different facets, but usually only taxonomic diversity is taken into account especially in poorly studied systems and for poorly studied groups such as those in this study. Considering all the facets of biodiversity and their drivers can be useful for conservation purpose, as such this kind of study deserve of publication. Nevertheless, I don't recommend for publishing in this form, but after a major revision. The sampling seems sufficient and the analyses appropriate, they also included some of the most relevant variables that could explain coleoptera and spider assembly. Nonetheless, the analyses should be better explained as there are many gaps in their description that make me concern about their appropriateness and did not make easier following the rationale behind the discussion. The interpretation of results and discussion should also be improved. I also think that the writing style should be restructured and English needs some revision.

The authors should consider the follow:

Introduction

Overall the introduction and hypothesis are not easy to follow and I found it bit confounding. I suggest restructure it and clarify the background. For example why is important analyse the different facets of biodiversity and their relationships? What determines the strength of correlation between TD, FD and PD? This is not clearly addressed.

L48 you are talking about diversity index so it is better to refer directly to phylogenetic and functional diversity here

L47-L54 these paragraphs could be joint and should be reworded

L55 some example of processes?

L57 FD reflect the differences of traits linking biodiversity, ecosystem function etc..

L60-61 yes but why it is important the understanding of their relationship? Explain more

L63-64 unclear, explain a bit more here

L65-66 not clear, Do you mean the amount of evolutionary history of a community? Please reword

L70 What means this sentence? What do you expect to see in phylogenetically conserved traits? What if they are not conserved? They will be shared/no shared among closely relates species? Please explain better

L75-77 Did you mean here that if TD correlate with PD or FD, they also correlate between them?

L80:82 yes, but why this is important for the studied taxa and ecosystem functioning?

L93 change stresses with stressors

L96 so the traits you considered are linked to them?

L103 I lack some information about the sensitivity of carabids and spider to environmental variables. For example, carabids are more affected by salinity, while spiders are more affected by e.g. vegetation? Given that you analyse the effect of environmental characteristics you should justify here why you are considering them

L109 Are you sure that Statzner in that paper said that?

L113-117 This is unclear and should be better addressed. Why salinity should effect the relationship between TD, PD and FD?

L118-119 This seems the contrary of what you said in the hypotheses 2. Probably you mean that you expect that environmental stressors influence carabids and spiders in the same directions?

L121 why recent?

Methods

L131 please remove the coordinates in the text. The figure is sufficient

L133 what is EUNIS A2? please explain and add a reference

L141 Please add a reference related to the tidal frequencies values

L149 Please removes the preservative liquid as you specify it later

L159 What did you mean here? Pitfalls were emptied after 4 and 12 days? If so why you have 29 and 48 continuous days of sampling: $5 \times 4 = 20$; $12 \times 5 = 60$. Please reword here as it is not clear

L172 remove the nearest %

L178:186 these are results. Please move them in the results section.

L202 Add reference used here

L203-204 please justifies the selected traits. Why they are important for Coleoptera and Spiders? Which are the ecological and functional significance of these traits in the community? How these traits can be affected by the environmental characteristics you are analysing?

Statistical analyses

Overall the analyses seem appropriate, but they should be better explained. The Bayesian approach HDI+ ROPE used is not clear, especially for readers not familiar with them. The same for the piecewise SEM. Please improve the clarity of the analyses used.

Moreover, it is not clear whether the authors are considering both salt marshes together or separately. I also suggest adding subsections for the different analyses done.

Taxonomic richness was calculated for trap, zone or wetland? Specify

Why you use variable in the SEM that not resulted significant with the ROPE? As far as I know, the HDI+ROPE is can be used for testing variable effects on the basis of the % of overlapping rope with CI, so why you are instead using the posterior distribution for considering which variable should be included in the model? This is not clear. For example salinity is significant for spider, but not for carabids but then you use salinity in the SEM mode. This part should be clarified.

L209 change estimates with estimator

L217-220 why you did not control for TD?

L226-230 I suggest here to explain a little how the HDI+ROPE decision rule works, for a better understanding

L230 Did you mean that you consider as significant variable those which percentage of Credible Interval is $> 95\%$? However, in the result you said that variables with 0% in ROPE are significant. This is confusing, especially to those unfamiliar with this analysis, please reword explaining better how this analyses works, or add the graphs as supplementary materials illustrating the % or ROPE that fall within the HDI, for a better understanding of the analyses used.

L237:240 Could you please provide a table with the initial and final model? Including the environmental variable included, as well the rational behind the initial model?

RESULTS

I suggest reorganizing the result in subsection according to their focus.

Please move here the lines from 178-186 regarding the environmental characteristics of the studied sites.

What about the taxonomic, phylogenetic and functional index?

L247 and L251 I guess you are expressing % of halophilic species considering the abundance, if so please add a graph or number in the related appendices, as this confounding. For example, from the appendix 3 I understand that 50% of the species are halophilic.

L248: the appendix 2 only mention which species is halophilic, but what about the others? I suggest adding a table with the selected traits of each species. Moreover in the appendix 1 you did not mention the tolerance to salinity as studied trait.

L260 What pd means ?Could you please explain this?

L287-302 please takes into account the direction of the interactions. Specify direct and indirect effects, which is why I guess you are using the SEM models
 L300 coefficient of standard estimate is 0.820, please correct

DISCUSSION

Several paragraph in the discussion are not related to the results obtained (e.g., L309-301; L322-324; L356-367, etc). They needs an important rewording, focusing only on objectives and results obtained.

L319 This seems more likely to explain the differences in the strengths of correlation between coleoptera and spider

L321 This was for the correlation between species richness and phylogenetic diversity

L 313 and 326 change link with correlation

L343 I guess you should point to your data and so refer to Appendix 5

L346 and what about spider?

L355 Could you please explain this?

L356-357 This sentence is not clear. Which diversity metrics are you referring here?

L357-359 this not explain your previous sentences

L363 but this is not your case as you have 6 halophilic species and six not halophilic specie

L356-367 I cannot understand the rationale behind this. You are talking about the stronger relationship between taxonomic richness and PD and FD, but you are not considering here the environmental drivers, which are analysed in the following paragraph. Moreover, as far as I know phylogenetic diversity can be also the result of biotic interaction, but this is never taken into account. I suggest introducing some sentences considering this.

L364 you never mention phylogenetic and functional structure in the results

L395-396 Could you please explain why litter depth affect FD?

L398 How can you affirm this? The correlation between FD and PD indicate that some unmeasured variable affected them but not that is the same variable.

Figure 2. Add correlation values

Decision letter (RSOS-202093.R0)

Dear Mr Ridel

The Editors assigned to your paper RSOS-202093 "Habitat filtering differentially modulates phylogenetic and functional diversity relationships between predatory arthropods" have now received comments from reviewers and would like you to revise the paper in accordance with the reviewer comments and any comments from the Editors. Please note this decision does not guarantee eventual acceptance.

Please submit your revised manuscript and required files (see below) no later than 21 days from today's (ie 26-Mar-2021) date. Note: the ScholarOne system will 'lock' if submission of the revision is attempted 21 or more days after the deadline. If you do not think you will be able to meet this deadline please contact the editorial office immediately.

on behalf of Professor Leslie Brown (Associate Editor) and Pete Smith (Subject Editor)
openscience@royalsociety.org

Associate Editor Comments to Author (Professor Leslie Brown):

Dear authors, thank you for submitting your paper to Royal Society Open Science for publication. We have received feedback from one reviewer and based on the comments received the manuscript needs major revision. I think that you have done research that are worthwhile publishing. Please read the reviewers' comments and attend to them. You must specifically pay attention to explain the statistical approach used since it makes it difficult for the reader to understand your interpretation of the results as well as the discussion. I also suggest you send the manuscript for language editing before you resubmit it. We hope that you will resubmit it for review after affecting the necessary changes.

Reviewer comments to Author:
Reviewer: 1
Comments to the Author(s)

This study examines the relationship between functional and phylogenetic diversity of two dominant ground dwelling arthropods in saltmarshes and the effect of environmental variables on them considering the influence of taxonomic diversity. This is a very interesting topic, as biodiversity has different facets, but usually only taxonomic diversity is taken into account especially in poorly studied systems and for poorly studied groups such as those in this study. Considering all the facets of biodiversity and their drivers can be useful for conservation purpose, as such this kind of study deserve of publication. Nevertheless, I don't recommend for publishing in this form, but after a major revision. The sampling seems sufficient and the analyses appropriate, they also included some of the most relevant variables that could explain coleoptera

and spider assembly. Nonetheless, the analyses should be better explained as there are many gaps in their description that make me concern about their appropriateness and did not make easier following the rationale behind the discussion. The interpretation of results and discussion should also be improved. I also think that the writing style should be restructured and English needs some revision.

The authors should consider the follow:

Introduction

Overall the introduction and hypothesis are not easy to follow and I found it bit confounding. I suggest restructure it and clarify the background. For example why is important analyse the different facets of biodiversity and their relationships? What determines the strength of correlation between TD, FD and PD? This is not clearly addressed.

L48 you are talking about diversity index so it is better to refer directly to phylogenetic and functional diversity here

L47-L54 these paragraphs could be joint and should be reworded

L55 some example of processes?

L57 FD reflect the differences of traits linking biodiversity, ecosystem function etc..

L60-61 yes but why it is important the understanding of their relationship? Explain more

L63-64 unclear, explain a bit more here

L65-66 not clear, Do you mean the amount of evolutionary history of a community? Please reword

L70 What means this sentence? What do you expect to see in phylogenetically conserved traits? What if they are not conserved? They will be shared/no shared among closely relates species? Please explain better

L75-77 Did you mean here that if TD correlate with PD or FD, they also correlate between them?

L80:82 yes, but why this is important for the studied taxa and ecosystem functioning?

L93 change stresses with stressors

L96 so the traits you considered are linked to them?

L103 I lack some information about the sensitivity of carabids and spider to environmental variables. For example, carabids are more affected by salinity, while spiders are more affected by e.g. vegetation? Given that you analyse the effect of environmental characteristics you should justify here why you are considering them

L109 Are you sure that Statzner in that paper said that?

L113-117 This is unclear and should be better addressed. Why salinity should effect the relationship between TD, PD and FD?

L118-119 This seems the contrary of what you said in the hypotheses 2. Probably you mean that you expect that environmental stressors influence carabids and spiders in the same directions?

L121 why recent?

Methods

L131 please remove the coordinates in the text. The figure is sufficient

L133 what is EUNIS A2? please explain and add a reference

L141 Please add a reference related to the tidal frequencies values

L149 Please removes the preservative liquid as you specify it later

L159 What did you mean here? Pitfalls were emptied after 4 and 12 days? If so why you have 29 and 48 continuous days of sampling: $5 \times 4 = 20$; $12 \times 5 = 60$. Please reword here as it is not clear

L172 remove the nearest %

L178:186 these are results. Please move them in the results section.

L202 Add reference used here

L203-204 please justifies the selected traits. Why they are important for Coleoptera and Spiders? Which are the ecological and functional significance of these traits in the community? How these traits can be affected by the environmental characteristics you are analysing?

Statistical analyses

Overall the analyses seem appropriate, but they should be better explained. The Bayesian approach HDI+ ROPE used is not clear, especially for readers not familiar with them. The same for the piecewise SEM. Please improve the clarity of the analyses used.

Moreover, it is not clear whether the authors are considering both salt marshes together or separately. I also suggest adding subsections for the different analyses done.

Taxonomic richness was calculated for trap, zone or wetland? Specify

Why you use variable in the SEM that not resulted significant with the ROPE? As far as I know, the HDI+ROPE is can be used for testing variable effects on the basis of the % of overlapping rope with CI, so why you are instead using the posterior distribution for considering which variable should be included in the model? This is not clear. For example salinity is significant for spider, but not for carabids but then you use salinity in the SEM mode. This part should be clarified.

L209 change estimates with estimator

L217-220 why you did not control for TD?

L226-230 I suggest here to explain a little how the HDI+ROPE decision rule works, for a better understanding

L230 Did you mean that you consider as significant variable those which percentage of Credible Interval is > 95%? However, in the result you said that variables with 0% in ROPE are significant. This is confusing, especially to those unfamiliar with this analysis, please reword explaining better how this analyses works, or add the graphs as supplementary materials illustrating the % or ROPE that fall within the HDI, for a better understanding of the analyses used.

L237:240 Could you please provide a table with the initial and final model? Including the environmental variable included, as well the rational behind the initial model?

RESULTS

I suggest reorganizing the result in subsection according to their focus.

Please move here the lines from 178-186 regarding the environmental characteristics of the studied sites.

What about the taxonomic, phylogenetic and functional index?

L247 and L251 I guess you are expressing % of halophilic species considering the abundance, if so please add a graph or number in the related appendices, as this confounding. For example, from the appendix 3 I understand that 50% of the species are halophilic.

L248: the appendix 2 only mention which species is halophilic, but what about the others? I suggest adding a table with the selected traits of each species. Moreover in the appendix 1 you did not mention the tolerance to salinity as studied trait.

L260 What pd means ?Could you please explain this?

L287-302 please takes into account the direction of the interactions. Specify direct and indirect effects, which is why I guess you are using the SEM models

L300 coefficient of standard estimate is 0.820, please correct

DISCUSSION

Several paragraph in the discussion are not related to the results obtained (e.g., L309-301; L322-324; L356-367, etc). They needs an important rewording, focusing only on objectives and results obtained.

L319 This seems more likely to explain the differences in the strengths of correlation between coleoptera and spider

L321 This was for the correlation between species richness and phylogenetic diversity

L 313 and 326 change link with correlation

L343 I guess you should point to your data and so refer to Appendix 5

L346 and what about spider?

L355 Could you please explain this?

L356-357 This sentence is not clear. Which diversity metrics are you referring here?

L357-359 this not explain your previous sentences

L363 but this is not your case as you have 6 halophilic species and six not halophilic specie

L356-367 I cannot understand the rationale behind this. You are talking about the stronger relationship between taxonomic richness and PD and FD, but you are not considering here the environmental drivers, which are analysed in the following paragraph. Moreover, as far as I know phylogenetic diversity can be also the result of biotic interaction, but this is never taken into account. I suggest introducing some sentences considering this.

L364 you never mention phylogenetic and functional structure in the results

L395-396 Could you please explain why litter depth affect FD?

L398 How can you affirm this? The correlation between FD and PD indicate that some unmeasured variable affected them but not that is the same variable.

Figure 2. Add correlation values

===PREPARING YOUR MANUSCRIPT===

===PREPARING YOUR REVISION IN SCHOLARONE===

Author's Response to Decision Letter for (RSOS-202093.R0)

See Appendix A.

Decision letter (RSOS-202093.R1)

Dear Mr Ridel

On behalf of the Editors, we are pleased to inform you that your Manuscript RSOS-202093.R1 "Habitat filtering differentially modulates phylogenetic and functional diversity relationships between predatory arthropods" has been accepted for publication in Royal Society Open Science subject to minor revision in accordance with the referees' reports. Please find the referees' comments along with any feedback from the Editors below my signature.

Please submit your revised manuscript and required files (see below) no later than 7 days from today's (ie 28-Apr-2021) date. Note: the ScholarOne system will 'lock' if submission of the revision is attempted 7 or more days after the deadline. If you do not think you will be able to meet this deadline please contact the editorial office immediately.

on behalf of Professor Leslie Brown (Associate Editor) and Pete Smith (Subject Editor)
openscience@royalsociety.org

Associate Editor Comments to Author (Professor Leslie Brown):
Associate Editor
Comments to the Author:

Thank you for the resubmission and detailed responses to the reviewers comments. The manuscript has been improved considerably and all the changes affected. I note that you had it

edited by a language editor, however there are still some language errors in terms of tenses and using plural vs single as well as sentences that does not make sense. I urge you to read through the manuscript in detail and to rectify these errors. I am listing only a few that I briefly picked up reading sections of it. There are many more, but I am just listing a few ones with different types of errors: e.g., Line 51: "It is therefore important to study both Taxonomic (TD), phylogenetic (PD), and functional (FD) diversities together for better" - you are listing three aspects TD, PD and FD and in the sentence you state "both" which refers to two - you should delete the word "both"; line 66: "However, it has been shown that assemblages with similar number of species can have different value of PD and/or FD " - "value" should be "values"; Line 103: "The risk of considering stress-related traits only is to study assemblages homogenized by the numerical dominance of salt-marsh adapted species, i.e. individual abundance of small, aeronaut, halophilic species (Pétillon et al. 2008)" - the sentence does not make sense and needs to be rephrased; Line 163: "Finally, a roof that is supported by metal stakes was set over the device to prevent the collecting liquid from water supply" - the sentence should be changes as follows "Finally, roof that is supported by metal stakes was placed over the device to prevent the collecting liquid from becoming diluted with water" - if that is not what you meant, it should be rephrased since it currently does not make sense.

I look forward to receiving the edited version.

===PREPARING YOUR MANUSCRIPT===

===PREPARING YOUR REVISION IN SCHOLARONE===

Author's Response to Decision Letter for (RSOS-202093.R1)

See Appendix B.

Decision letter (RSOS-202093.R2)

Dear Mr Ridel,

It is a pleasure to accept your manuscript entitled "Habitat filtering differentially modulates phylogenetic and functional diversity relationships between predatory arthropods" in its current form for publication in Royal Society Open Science. The comments of the reviewer(s) who reviewed your manuscript are included at the foot of this letter.

on behalf of Professor Leslie Brown (Associate Editor) and Pete Smith (Subject Editor)
openscience@royalsociety.org

Associate Editor Comments to Author (Professor Leslie Brown):

Comments to the Author:

Thanks you for all the changes affected to the manuscript as requested.

Appendix A

Dear Editor,

We would like to submit the revised version of our manuscript entitled “Habitat filtering differentially modulates phylogenetic and functional diversity relationships between predatory arthropods”.

We have carefully addressed all the points raised by the reviewer (see the detailed response below) and paid for a professional editing of English, which resulted in an improved version of our work (see acknowledgments of the revised ms).

Thank you for considering our revised manuscript.

Yours sincerely,

Aurélien Ridel, corresponding author

This study examines the relationship between functional and phylogenetic diversity of two dominant ground dwelling arthropods in saltmarshes and the effect of environmental variables on them considering the influence of taxonomic diversity. This is a very interesting topic, as biodiversity has different facets, but usually only taxonomic diversity is taken into account especially in poorly studied systems and for poorly studied groups such as those in this study. Considering all the facets of biodiversity and their drivers can be useful for conservation purpose, as such this kind of study deserve of publication.

Thanks for this global overview of our work, and for all the comments and suggestions that were highly appreciated (see L455).

Nevertheless, I don't recommend for publishing in this form, but after a major revision. The sampling seems sufficient and the analyses appropriate, they also included some of the most relevant variables that could explain coleoptera and spider assembly. Nonetheless, the analyses should be better explained as there are many gaps in their description that make me concern about their appropriateness and did not make easier following the rationale behind the discussion. The interpretation of results and discussion should also be improved. I also think that the writing style should be restructured and English needs some revision. The authors should consider the follow:

Introduction

Overall the introduction and hypothesis are not easy to follow and I found it bit confounding. I suggest restructure it and clarify the background. For example why is important analyse the different facets of biodiversity and their relationships? What determines the strength of correlation between TD, FD and PD? This is not clearly addressed.

The introduction has been reworked on all points suggested by the reviewer to clarify the importance of analysing all facets of biodiversity (taxonomic, phylogenetic and functional) together, and to clarify which parameters influence the strength of correlation between this diversity metrics (see e.g. L47 and L126).

L48 you are talking about diversity index so it is better to refer directly to phylogenetic and functional diversity here

We reworded this paragraph to refer directly to phylogenetic and functional diversity as suggested (L47-54):

“This approach does not consider all facets of biodiversity, such as accumulated evolutionary history traits that can be highlighted through the phylogenetic diversity (Webb et al., 2002) or the diversity of morphological, physiological, and ecological traits of an assemblage that can be revealed by functional diversity (Petchey and Gaston, 2002, 2006). It is therefore important to study both Taxonomic (TD), phylogenetic (PD), and functional (FD) diversities together for better understanding of the composition and dynamics of species assemblages (Webb et al., 2002), or even to set up priorities for biodiversity conservation in a fairer way (Swenson, 2011).”

L47-L54 these paragraphs could be joint and should be reworded

As indicated above these paragraphs have been merged and rewritten.

L55 some example of processes?

Changed accordingly (L56).

L57 FD reflect the differences of traits linking biodiversity, ecosystem function etc..

Changed accordingly (L58).

L60-61 yes but why it is important the understanding of their relationship? Explain more

We completed the sentence as follow (L63-64):

“Despite the fact that these metrics are seen to be complementary, their mutual relationships remain unclear (Devictor et al., 2010), yet studying them is necessary to better understand all forces driving biodiversity patterns.”

L63-64 unclear, explain a bit more here

We reworded the sentence as follow (L66-67):

“However, it has been shown that assemblages with similar number of species can have different value of PD and/or FD (Safi *et al.*, 2011; Tucker and Cadotte, 2013).”

L65-66 not clear, Do you mean the amount of evolutionary history of a community? Please re-word

Yes, and we clarified the sentence as follow (L68-69):

“Moreover, the strength of a correlation between TD and PD depends on the time of evolutionary history of a given community,”

L70 What means this sentence? What do you expect to see in phylogenetically conserved traits? What if they are not conserved? They will be shared/no shared among closely relates species? Please explain better

To improve comprehension, we completed the sentence as follow (L74-75):

“If traits are phylogenetically conserved, PD can also provide information about unmeasured functional traits (Cadotte et al., 2009), because in this case, PD results of the addition of all functional changes that occurred in the past.”

L75-77 Did you mean here that if TD correlate with PD or FD, they also correlate between them?

We meant that the fact taxonomic diversity is part of the formula for phylogenetic and functional diversity calculations creates a mathematical correlation artefact. We reworded the sentence to make it clearer (L79-81):

“Finally, the inclusion of TD in both PD and FD calculations can lead to a correlation between them due to a mathematical correlation artefact caused by a side effect”

L80:82 yes, but why this is important for the studied taxa and ecosystem functioning?

We completed the sentence as follow (L84-87):

“To have a better understanding of the relationships between diversity metrics, it is important to understand what the drivers of these metrics are and how they affect the relationships between them. Moreover, highlighting these drivers can improve the understanding of ecosystem functioning, as well as the observed biodiversity patterns, across all components of biodiversity.”

L93 change stresses with stressors

Changed accordingly (L99).

L96 so the traits you considered are linked to them?

Not fully, because some of these traits are indeed related to salt-marsh habitats (e.g. ballooning propensity and salt-marsh recolonization after flood), some are not directly (e.g. body size or hunting guilds) and finally several traits important for living in salt marshes are not considered here (e.g. osmoregulation abilities). We completed the sentence as follow (L100-106):

“These stresses have a strong impact on salt-marsh organisms (Lefeuvre et al., 2003), and most of the species found in these ecosystems have a high phenotypic plasticity, or even morphological, physiological, or behavioural adaptations to cope with the stresses (Pennings & Callaway, 1992; Pétilion et al., 2009, 2011). The risk of considering stress-related traits only is to study assemblages homogenized by the numerical dominance of salt-marsh adapted species, i.e. individual abundance of small, aeronaut, halophilic species (Pétilion et al. 2008).”

L103 I lack some information about the sensitivity of carabids and spider to environmental variables. For example, carabids are more affected by salinity, while spiders are more affected by e.g. vegetation? Given that you analyse the effect of environmental characteristics you should justify here why you are considering them

We agree and we added some details of sensibility of each taxon in the second hypothesis (see response below and L127-129).

L109 Are you sure that Statzner in that paper said that?

Thanks for this comment, we replaced (Statzner *et al.*, 2001) by (Thienemann 1918, 1920 cited by Statzner *et al.*, 2001) (L118).

L113-117 This is unclear and should be better addressed. Why salinity should effect the relationship between TD, PD and FD?

Because salinity could select species with similar functional characteristics, we modified the sentence as follow (L122-129):

“Hypothesis 2: Despite the fact that TD influences the strength of the correlation between PD and FD by side effects (Safi *et al.*, 2011; Pavoine *et al.*, 2013; Cadotte *et al.*, 2019), we expect a relationship between TD and both PD and FD diversities stronger for carabids due to the greater sensitivity of carabids to environmental constraints such as salinity (Pétilion *et al.*, 2008), resulting in a pool of species with close functional traits for e.g. resisting salinity, avoiding flooding and/or recolonizing the marsh after tides. One the way around, we expect spider assemblages to be more driven by changes in vegetation structure which has been shown in coastal (Hacala et al. 2020) as well as in other inland habitats (see e.g. Lafage et al. 2019).”

L118-119 This seems the contrary of what you said in the hypotheses 2. Probably you mean

that you expect that environmental stressors influence carabids and spiders in the same directions?

Yes, and we changed the sentence accordingly (L130).

L121 why recent? Methods

We added a justification in the sentence as follow (L132-134):

“In addition, the environmental factors should influence the PD of salt-marsh organisms similarly because of their recent (less than 6,000 years) (Desender *et al.*, 1998) evolutionary history in salt marshes”

L131 please remove the coordinates in the text. The figure is sufficient

The coordinates have been removed (L142-143).

L133 what is EUNIS A2? please explain and add a reference

We completed as follow (L145-146):

“(EUNIS A2.5, Coastal saltmarshes and saline reedbeds, according to the European Nature Information System habitats typology)”

L141 Please add a reference related to the tidal frequencies values

We completed as follow (L154-156):

“The high marshes have about 0.5% of annual recovering tides whereas the low marshes have about 60% (based on field observations by the managers of both reserves).”

L149 Please removes the preservative liquid as you specify it later

Changed accordingly (L161).

L159 What did you mean here? Pitfalls were emptied after 4 and 12 days? If so why you have 29 and 48 continuous days of sampling: $5 \times 4 = 20$; $12 \times 5 = 60$. Please reword here as it is not clear

To clarify the protocol, we reworded the sentence (L173-175):

“The dates and duration of the sampling sessions differed between sites, along with differences in immersion frequencies. All sessions accumulated, 29 days of sampling were done on the first site, and 48 days on the second site.”

L172 remove the nearest %

Changed accordingly (L184).

L178:186 these are results. Please move them in the results section.

Changed accordingly (L261-270).

L202 Add reference used here

References were provided in the appendix, and we changed the sentence accordingly (L203-205):

“In order to calculate the functional diversity per pitfall trap for each taxon, functional traits were assigned to each of the spider and carabid species according to the literature cited in Appendix 1.”

L203-204 please justifies the selected traits. Why they are important for Coleoptera and Spiders? Which are the ecological and functional significance of these traits in the community? How these traits can be affected by the environmental characteristics you are analysing?

We completed the paragraph as follow (L205-209):

“The selected traits, viz. size, dispersal capacity and overall diet, were chosen to be 1) relevant traits for computing FD of each group, and 2) comparable between the two taxa. Long-distance dispersal, body size and overall diet were chosen to broadly reflect species colonization ability, stress tolerance capacities and diversity of prey resources, respectively (see e.g. Hacala et al., in revision)”

Statistical analyses

Overall the analyses seem appropriate, but they should be better explained. The Bayesian approach HDI+ ROPE used is not clear, especially for readers not familiar with them. The same for the piecewise SEM. Please improve the clarity of the analyses used. Moreover, it is not clear whether the authors are considering both salt marshes together or separately. I also suggest adding subsections for the different analyses done. Taxonomic richness was calculated for trap, zone or wetland? Specify

We completed the sentence as follow (L212-213):

“Taxonomic diversity was estimated by measuring the species richness of samples per trap (both sites together), and computed using the BAT package”

Why you use variable in the SEM that not resulted significant with the ROPE? As far as I know, the HDI+ROPE is can be used for testing variable effects on the basis of the % of overlapping rope with CI, so why you are instead using the posterior distribution for considering which variable should be included in the model? This is not clear. For example salinity is significant for spider, but not for carabids but then you use salinity in the SEM mode. This part should be clarified.

We developed this part accordingly (see below and L235-243).

L209 change estimates with estimator

Changed accordingly (L214).

L217-220 why you did not control for TD?

We did not control for TD because one of our goals was to show, with the SEM, that using only correlation between PD and FD can potentially lead to wrong conclusions.

L226-230 I suggest here to explain a little how the HDI+ROPE decision rule works, for a better understanding

We developed here to clarify (L235-243):

“This rule states that if the HDI is completely outside the ROPE, one can reject the “null hypothesis” for the particular parameter. Symmetrically, if the ROPE completely covers the HDI, one can accept the null hypothesis. Otherwise, whether to accept or reject the null hypothesis remains undecided. Variables were selected as candidates for the Structural equation models when ROPE > 95% which means we accepted variables for which we could reject the null hypothesis and variables for which we could not decide whether or not to reject the null hypothesis (under the limit of 5%). We also provide we the probability of direction (pd), which is the probability that the posterior distribution of a parameter is strictly positive or negative.”

L230 Did you mean that you consider as significant variable those which percentage of Credible Interval is > 95%? However, in the result you said that variables with 0% in ROPE are significant. This is confusing, especially to those unfamiliar with this analysis, please reword explaining better how this analyses works, or add the graphs as supplementary

materials illustrating the % or ROPE that fall within the HDI, for a better understanding of the analyses used.

As the reviewer noticed correctly from the result section, only variables with 0% in ROPE were considered significant. Nevertheless, when the ROPE does not completely cover the HDI, the variables significance remains undecided. In order to avoid excluding variables that could possibly have an effect we allowed variables with up to 5% in ROPE to be included in the SEM model. Precisions were added in the method section (L235-243).

L237:240 Could you please provide a table with the initial and final model? Including the environmental variable included, as well the rational behind the initial model?

Thanks for this suggestion, we added a new table as follow:

Table 2. Initial and final SEM models for spiders and carabid beetles. All models included site/zone as a random factor and correlated error between PD and FD. PD: phylogenetic diversity, FD: functional diversity, TD: taxonomic diversity.

Spiders	Initial model	PD ~ Salinity + Vegetation Height + TD FD ~ PD + Litter depth + TD TD ~ Vegetation Height
	Final model	PD ~ Salinity + TD FD ~ PD + Litter depth + TD
Carabids	Initial model	PD ~ Salinity + Vegetation Height + TD
		FD ~ PD + Vegetation Height + TD
		PD ~ Salinity + Vegetation Height
	Final model	PD ~ TD
		FD ~ PD + TD
TD ~ Salinity		

RESULTS

I suggest reorganizing the result in subsection according to their focus. Please move here the lines from 178-186 regarding the environmental characteristics of the studied sites.

Thanks for this relevant comment, we added title for all subsections (see L261, L272, L282, L284, L318).

What about the taxonomic, phylogenetic and functional index?

A table containing the values obtained for each site and both taxa has been added as an appendix.

Appendix 6. Values of taxonomic (TD) functional (FD) and phylogenetic (FD) diversities, for each site (FA= fier d'Ars, MO = Moëze Brouage) and for each taxon (spid= spiders, car= carabids) with associated standard deviation.

site	FA	MO
TD spiders	11.52 ± 8.84	19.56 +-6.25
FD spiders	2.90 +- 1.84	4.43 +-1.3
PD spiders	44.64 +- 20.72	42.48 +- 14.81
TD carabids	3.08 +- 3.14	3.77 +- 2.86
FD carabids	1.34 +- 1.42	1.46 +- 0.96
PD carabids	7.68 +- 8.03	8.69 +- 4.96

L247 and L251 I guess you are expressing % of halophilic species considering the abundance, if so please add a graph or number in the related appendices, as this confounding. For example, from the appendix 3 I understand that 50% of the species are halophilic.

Yes, the text is referring to the number of individuals (see L273) while the appendix mentions whether the species are considered halophilic or not (L780).

L248: the appendix 2 only mention which species is halophilic, but what about the others? I suggest adding a table with the selected traits of each species. Moreover, in the appendix 1 you did not mention the tolerance to salinity as studied trait.

In appendices 2 and 3 we added “the other are considered non-halophilic”. Moreover, according to your suggestion, we added all values of functional traits of all species sampled. In Appendix 1 we did not mention the tolerance to salinity as studied traits because it is not considered a functional trait as such, but more a consequence of several morphological, behavioural or physiological characteristics. Thanks to previous analyses not included here, we also know that include either halophily or these adaptations to salt-marsh habitats is not discriminating in functional analyses because the environmental filtering is so strong that the majority of individuals (but not species) are halophilic.

L260 What pd means? Could you please explain this?

We developed this part accordingly (L241-243):

“We also provide we the probability of direction (pd), which is the probability that the posterior distribution of a parameter is strictly positive or negative.”

L287-302 please takes into account the direction of the interactions. Specify direct and indirect effects, which is why I guess you are using the SEM models

Changed accordingly (L322-324), as follow:

“Salinity and litter depth were linked to spider PD (positive link, selected in the model but only marginally significant: $p = 0.068$) and FD (negative link, selected in the model and again almost significant: $p = 0.051$), respectively”

L300 coefficient of standard estimate is 0.820, please correct

Changed accordingly (L332).

DISCUSSION

Several paragraph in the discussion are not related to the results obtained (e.g., L309-311; L322-324; L356-367, etc). They needs an important rewording, focusing only on objectives and results

We followed this recommendation by removing these parts (L340 and L251), and by rewording the third one (see below and L387-397).

L319 This seems more likely to explain the differences in the strengths of correlation between coleoptera and spider

We developed this part accordingly (see below and L347-351).

L321 This was for the correlation between species richness and phylogenetic diversity

No, a reduced number of species can increase the correlation between PD and FD as in mention in Tucker and Cadotte 2013, regardless of correlation with TD. To clarify we developed as follow (L347-351):

“In addition, the relatively small number of species collected in the studied salt marshes (N=55 for spiders and N=12 for carabids) compared to that in less constrained environments from the same biogeographic area (e.g. see Lafage et al. 2015: N=99 for spiders and N=43 for carabids) may also increase the strength of the PD-FD correlation (Tucker and Cadotte 2013), especially for carabids.”

L 313 and 326 change link with correlation
Changed accordingly (L353).

L343 I guess you should point to your data and so refer to Appendix 5
The reference to Appendix 5 (as well as a reference to appendix 4) were added accordingly (L368 and L371).

L346 and what about spider?

We modified the sentence as follow (L372-374):

“Therefore, the observed relationship between phylogenetic and functional diversities suggests that functional traits are phylogenetically more conserved in carabids than in spiders”

L355 Could you please explain this?

We developed to improve comprehension (L379-385):

“These results suggest that the relationship between phylogenetic and functional diversities is mainly affected by a side effect due to the inclusion of taxonomic diversity in both phylogenetic and functional diversities calculation which corroborates with previous studies. This correlation is usually increasing with a decreasing number of species (Tucker and Cadotte, 2013)., which could also explain the pattern we observed in carabids (low TD, high R²) vs spider (higher TD, lower R²).”

L356-357 This sentence is not clear. Which diversity metrics are you referring here?

Here, in “other diversity metrics” we meant phylogenetic and functional diversities. We completed as follow (L387-388):

“Expectedly, taxonomic diversity was also found to be more strongly related to functional and phylogenetic diversities in carabids than in spiders.”

L357-359 this not explain your previous sentences

Indeed, but this sentence is to be connected with the next ones. Because of carabid larval stages are considered more sensitive to salinity (edaphic larvae), this taxon is probably more adapted to this constraint. And the species pool of carabids is indeed composed by a large majority of species phylogenetically and functionally close, which increases the correlation between taxonomic diversity and other diversity metrics. We decided to rewrite all the concerned paragraph (see L387-397, and also comment below):

L363 but this is not your case as you have 6 halophilic species and six not halophilic specie
Thanks for this comment, we meant individuals not species, and the sentence was changed accordingly (391-392).

L356-367 I cannot understand the rationale behind this. You are talking about the stronger relationship between taxonomic richness and PD and FD, but you are not considering here the environmental drivers, which are analysed in the following paragraph.

We wanted to discuss on the effect of taxonomic diversity on PD-FD relationship alone first, and developed the effect of environmental drivers on TD PD and FD later (in the next paragraph “Effects of environmental filtering on diversity metrics”).

Moreover, as far as I know phylogenetic diversity can be also the result of biotic interaction, but this is never taken into account. I suggest introducing some sentences considering this.

Thanks for this comment, but as also stated by this referee, it is important to stick on the results we had and this study actually considered environmental factors only.

Moreover, in accordance with the previous comments, we have reworded the whole paragraph “Effects of taxonomic diversity on PD-FD relationships” as follow (L387-397):

“Expectedly, taxonomic diversity was also found to be more strongly related to functional and phylogenetic diversities in carabids than in spiders. Because of their ecology (larval stage in the soil), carabids are more exposed to salinity than spider and possess particular morphological adaptive features such as a waterproof-like inter-tegument cuticle (Foster & Treherne, 1976). This resulting in a specialised halophilic species pool for this taxa (99.7% of individuals are halophilic, in opposition to 58.9% of halophilic individuals for spiders). Furthermore, as stated above, halophilic carabids seem to be more phylogenetically clustered than halophilic spiders (Appendix 4 and 5), resulting in a strong link between the taxonomic diversity and both phylogenetic and functional diversities for the taxa. The lower percentage of halophilic individuals (58.9%) compared to that of halophilic carabid individuals (99.7%) obtained in this study is consistent with our hypothesis.”

L364 you never mention phylogenetic and functional structure in the results

We added a reference to appendices 4 and 5 to explain this phylogenetic clustering (L394).

L395-396 Could you please explain why litter depth affect FD?

We completed the sentence as follow (L424-427):

“The effect of litter depth on spider functional diversity could be explained by a change in hunting guilds driven by a modification of prey composition with litter depth (see also Uetz, 1979 and Döbel et al., 1990)”

L398 How can you affirm this? The correlation between FD and PD indicate that some unmeasured variable affected them but not that is the same variable.

We completed the sentence as follow (L427-430):

“It is important to point out that the unidentified environmental variable affecting the functional diversity of carabids is the same as the one driving their phylogenetic diversity (as indicated by the strong correlated error found between the two response variables), thus strengthening the links between these two metrics”

Figure 2. Add correlation values

Changed accordingly (L752).

Appendix B

Dear Editor,

We would like to submit the new revised version of our manuscript entitled “Habitat filtering differentially modulates phylogenetic and functional diversity relationships between predatory arthropods”.

An extensive proofreading of the English language has been carried out, and the errors pointed out by the reviewer have been corrected.

Thank you for the resubmission and detailed responses to the reviewer’s comments. The manuscript has been improved considerably and all the changes affected.

Thank you for your constructive opinions which helped to improve the manuscript

I note that you had it edited by a language editor, however there are still some language errors in terms of tenses and using plural vs single as well as sentences that does not make sense. I urge you to read through the manuscript in detail and to rectify these errors. I am listing only a few that I briefly picked up reading sections of it. There are many more, but I am just listing a few ones with different types of errors:

we are surprised by these errors following professional proofreading, thank you for pointing them out. We have again carried out a full proofreading of the English by the last author who lived for a while in USA (and indeed we found new typos, some resulting from tracked changes).

e.g., Line 51: "It is therefore important to study both Taxonomic (TD), phylogenetic (PD), and functional (FD) diversities together for better" - you are listing three aspects TD, PD and FD and in the sentence you state "both" which refers to two - you should delete the word "both"

Changed accordingly

line 66: "However, it has been shown that assemblages with similar number of species can have different value of PD and/or FD " - "value" should be "values"

Changed accordingly

Line 103: "The risk of considering stress-related traits only is to study assemblages homogenized by the numerical dominance of salt-marsh adapted species, i.e. individual abundance of small, aeronaut, halophilic species (Pétillon et al. 2008)" - the sentence does not make sense and needs to be rephrased

We changed by “The risk of considering only functional traits related to salt-marsh stressors is to study assemblages homogenized by the dominance of salt-marsh adapted species, i.e. high abundance of small, aeronaut, halophilic species (Pétillon et al. 2008)” and moved line 214.

Line 163: "Finally, a roof that is supported by metal stakes was set over the device to prevent the collecting liquid from water supply" - the sentence should be changes as follows "Finally, roof that is supported by metal stakes was placed over the device to prevent the collecting liquid from becoming diluted with water" - if that is not what you meant, it should be rephrased since it currently does not make sense.

Changed accordingly

I look forward to receiving the edited version.